# Enhancing the Nutritional Value of Red Meat through Genetic and Feeding Strategies

**DOI:** 10.3390/foods10040872

**Published:** 2021-04-16

**Authors:** Manuel Juárez, Stephanie Lam, Benjamin M. Bohrer, Michael E. R. Dugan, Payam Vahmani, Jennifer Aalhus, Ana Juárez, Oscar López-Campos, Nuria Prieto, Jose Segura

**Affiliations:** 1Lacombe Research and Development Centre, Agriculture and Agri-Food Canada, Lacombe, AB T4L 1W1, Canada; stephanie.lam@canada.ca (S.L.); mike.dugan@canada.ca (M.E.R.D.); Jennifer.Aalhus@outlook.com (J.A.); oscar.lopezcampos@canada.ca (O.L.-C.); Nuria.PrietoBenavides@canada.ca (N.P.); jose.seguraplaza@canada.ca (J.S.); 2Department of Animal Sciences, The Ohio State University, Columbus, OH 43210, USA; bohrer.13@osu.edu; 3Department of Animal Science, University of California Davis, Davis, CA 95616, USA; pvahmani@ucdavis.edu; 4Oceanographic Center of Cadiz, Spanish Institute of Oceanography (IEO), 11006 Cádiz, Spain; ana.juarez@ieo.es

**Keywords:** beef, lamb, pork, trace elements, micronutrients, fatty acids, genomics, heritability

## Abstract

Consumption of red meat contributes to the intake of many essential nutrients in the human diet including protein, essential fatty acids, and several vitamins and trace minerals, with high iron content, particularly in meats with high myoglobin content. Demand for red meat continues to increase worldwide, particularly in developing countries where food nutrient density is a concern. Dietary and genetic manipulation of livestock can influence the nutritional value of meat products, providing opportunities to enhance the nutritional value of meat. Studies have demonstrated that changes in livestock nutrition and breeding strategies can alter the nutritional value of red meat. Traditional breeding strategies, such as genetic selection, have influenced multiple carcass and meat quality attributes relevant to the nutritional value of meat including muscle and fat deposition. However, limited studies have combined both genetic and nutritional approaches. Future studies aiming to manipulate the composition of fresh meat should aim to balance potential impacts on product quality and consumer perception. Furthermore, the rapidly emerging fields of phenomics, nutrigenomics, and integrative approaches, such as livestock precision farming and systems biology, may help better understand the opportunities to improve the nutritional value of meat under both experimental and commercial conditions.

## 1. Introduction

Meat consumption has played a substantial role in human evolution. The ability of early humans to use basic rudimentary tools, such as stones and sticks to control and use fire to cook meat products procured from hunting, as well as their access to bone marrow, led to the consumption of nutrient dense foods with higher energy and protein content. This lessened the need for large jaws/teeth and a bulky digestive system and consequently, resulted in a larger endocranial cavity allowing for brain growth to occur [1,2]. In regions of the world where humans have thrived, animal products, especially meat products, have been an important component in the human diet due to its high level of biologically available nutrients, including protein, iron, zinc and B-complex vitamins, especially B_12_ [3,4].

Despite the importance of red meat in human evolution and its high nutritional value in the human diet, the consumption of red meat (defined in this review as livestock meat from beef, pork and lamb) has received significant negative attention in both scientific studies and the popular press. This includes the increasing number of studies suggesting the association between red meat consumption and negative effects on human health [5,6,7] and the environment [8,9]. However, human dietary studies often have no supporting evidence of a cause-and-effect relationship between red meat consumption and negative effects on human health. Furthermore, factors other than red meat consumption, including meat processing and cooking methods, lifestyle factors [10,11,12], and adiposity index of the individuals [13] are known to influence negative effects attributed to red meat consumption, such as inflammation. However, as studies emerge, more evidence is revealing that the removal of red meat from the human diet may lead to negative health effects, such as lower bone mineral density and higher bone fracture rates [14]. In support, studies have shown the importance of red meat nutrients for the pregnant and aging population to maintain healthy vitamin and mineral status and skeletal muscle mass, respectively [15]. Nevertheless, negative attention towards the consumption of red meat is increasing, especially the perception of a strong association between the consumption of red meat products and the development of some forms of cancer [16]. Parallel concerns on the consumption of red meat and its impact on climate change and greenhouse gas emissions for most nations in the world is also increasing [8,9], leading to an emphasis on improving the sustainability of livestock production. It is known that diets high in refined foods, sugars, oils and meats are expected to contribute up to 80% of the increase in greenhouse gases originating from food production and global land clearing [9]. In addition to environmental concerns, animal welfare, food safety, and cost of meat production are becoming increasingly important in developed countries [17,18]. As a consequence, new recommendations from national and international health organizations in developed countries suggest a decrease in the consumption of the protein food group, including red meats, and an increase in the consumption of the fruits and vegetables food group which includes plant-based protein alternatives [19,20].

In recent years, a constant decrease in per capita meat consumption in developed countries has been observed; despite this, meat remains a key component of human diets around the world [21], with more than 90% of consumers regularly eating meat in developed countries [22,23]. Following this trend, meat consumption in developing countries continues to increase every year, with projections indicating a parallel increase alongside the growth in world population [24]. With different demands and increases in red meat consumption worldwide, the variability in dietary recommendations that exists across the world in different regions should also be considered, as these dietary recommendations are specific to different societal and cultural norms [25]. This has shown that different regions worldwide have varying recommendations for portions of the food groups and what is considered ’healthy’ has changed over time [25]. This has emphasized the importance of understanding the potential to manipulate red meat nutrients to meet specific needs of different regions. The nutrient density and nutrient value of meat products has been reviewed and measured, showing the highly bioavailable and nutrient dense quality of beef, pork and lamb compared to other non-meat food products [3]. However, the understanding of what environmental and genetic factors influence the variability in the nutrient composition of red meat is incomplete.

Variability in the nutrient composition of red meat is known to exist due to differences in animal species (ruminant and monogastric) and breeds, as well as different strategies for feeding and rearing livestock, which has allowed for opportunities to manipulate the nutritional value. This has been studied by implementing different strategies to enhance its beneficial properties [26,27,28], resulting in value-added meats with enhanced composition which are now available in the marketplace. For example, research studies have focused on docosapentaenoic acid (DPA) in red meat as a valuable terrestrial source of long chain fatty acids, to better understand the value and content of naturally occurring *trans* fatty acids present in red meat [29]. In support, numerous studies have provided evidence that nutrient composition variation in red meat depends on the complex interaction between the animal’s genetics, the production environment and their interaction. While environmental factors can influence meat nutritional value, such as production system, animal age, gender, or physical activity, diet has shown to have the largest impact on red meat composition and will be the main focus of this review, along with genetic studies including both breed comparisons and genetic parameter analyses.

## 2. Literature Review Methodology

A literature search was conducted using the following databases: Scopus, Biological Abstracts, CAB Abstracts and FSTA. The search terms included: ’beef’ AND/OR ’bovine’ AND/OR ’pork’ AND/OR ’swine’ AND/OR ’lamb’ AND/OR ’ovine’ AND/OR ’red meat’ AND/OR ’*longissimus thoracis lumborum*; muscle’ AND/OR ’REA’ AND/OR ’ribeye’ AND ’nutrition’ AND/OR ’diet’ AND/OR ’supplement’ AND/OR ’additive’ AND/OR ’feeding program’ AND/OR ’genetic’ AND/OR ’heritability’ AND/OR ’genomic breeding values’ AND/OR ’GEBVs’ AND ’minerals’AND/OR ’trace elements’ AND/OR ’copper’ AND/OR ’iron’ AND/OR ’zinc’ AND/OR ’selenium’ AND/OR ’iodine’ AND/OR ’sodium’ AND/OR ’calcium’ AND/OR ’cobalt’ AND/OR ’vitamin A’ AND/OR ’vitamin E’ AND/OR ’vitamin B_2_’ AND/OR ’vitamin B_9_’ AND/OR ’vitamin B_12_’ AND/OR ’fat’ AND/OR ’fatty acids’ AND/OR ’*trans*’ AND/OR ’conjugated linoleic acids’. The search was filtered for publication dates during or after the year 1999 and excluded documents on processed meat, human nutrition, and clinical trials. The results (6800) were sorted by species and nutrient. In addition, the references from the articles obtained by this method were used to identify additional relevant material.

## 3. Manipulating the Nutritional Value of Red Meat

Dietary nutrients are categorized as micro- or macronutrients based on dietary requirements of humans. Micronutrients are further categorized into vitamins and trace elements. Water soluble vitamins include B-complex vitamins, while fat soluble vitamins include vitamins A, D, and E, among others. Trace elements can be grouped between those highly influenced by diet and liver metabolism, such as iron, copper, and zinc, and those less affected, such as iodine and selenium. Macronutrients include proteins and fats, as well as carbohydrates and water; however, the latter two are not as relevant when considering the nutritional value of red meat and therefore are not included in this review.

### 3.1. Micronutrients

#### 3.1.1. Vitamins

Concentration ranges for several vitamins in beef, pork and lamb are shown in Table 1, revealing a high variability in the concentration of vitamins in red meat among different species.

Although retinoic acid and retinal are the most physiologically active forms of vitamin A, other forms also include free retinol, retinyl esters, β-carotene, and carotenals. Vitamin A is essential for maintaining normal vision, the immune system and reproductive function. Vitamin A deficiency is common in undeveloped countries, especially among children and women at reproductive age, but is rarely seen in more developed countries [30]. It is mainly stored in the liver, but several studies have also shown potential for manipulation of its content in milk, plasma and subcutaneous fat through dietary supplementation of livestock [31,32,33].

Domínguez et al. [33] studied the effect of feeding chestnuts to Celta pigs during the finishing phase on retinol concentration in adipose tissue. Their results show a retinol concentrations ranging from 0.63 to 0.76 μg/g retinol in various tissue including rump fat, subcutaneous *biceps femoris*, and subcutaneous dorsal fat, and a concentration of 527 μg/g in the liver. Results in beef *longissimus* from Duckett et al. [34] showed a greater concentration in α-tocopherol, β-carotene, thiamine and riboflavin in cattle managed in a pasture-finishing system compared to counterparts managed in a high concentrate-finishing system. In addition, a study comparing different lamb rearing systems by Osorio et al. [35] revealed a highly significant difference between retinol concentration in lamb *longissimus* muscle in maternal milk rearing (10.83 μg/100 g) compared to milk replacement rearing (43.69 μg/100 g). Similarly, Blanco et al. [36] measured vitamin concentration in *longissimus thoracis* muscle of lambs and compared production systems during rearing of suckling lambs raised indoors to suckling lambs raised on pasture. The results reveal a highly significant difference in lutein, retinol, α-tocopherol, and γ-tocopherol, with higher concentrations observed in *longissimus* muscle of suckling lambs in the pasture-raised system. Taylor et al. [37] described the first heritability values (0.36, 0.03, 0.79 and 0.48) related to hepatic vitamin A concentration in beef cattle at 235, 340, 600 and 710 days of age, respectively. More recently, Kato et al. [38] also described decreasing heritability values with age of 0.37, 0.24, 0.16 and 0.07 (at <13, 14–18, 19–21 and >13 months, respectively) for vitamin A concentration in serum from calves with Japanese Black sires and Holstein dams. Nevertheless, the potential for genetic manipulation by assessing genetic parameters and heritability of vitamin A in livestock muscle has not been reported to date.

Vitamin E deficiency is caused by inadequate dietary intake or by a disorder causing fat malabsorption. Due to its important role, vitamin E is a common ingredient supplemented in animal nutrition [39]. With its importance in overall health to consumers as well as its role in shelf-life stability, there is interest to increase vitamin E concentration in red meat. Multiple studies have shown the potential to increase vitamin E in different red meat products, using it as a biological antioxidant to enhance meat color stability and slow the rate of lipid oxidation [40]. Absorption of vitamin E is known to be proportional to the vitamin E status of the animal in most species and this status varies by several factors, including fat intake, digestion, liver function or an excess of dietary zinc. Muscle deposition mainly depends on dietary compound source (synthetic, *all-rac*-α-tocopheryl acetate vs. natural, RRR-α-tocopheryl acetate), dosage, species, tissue and time of supplementation [41]. Leal et al. [42] described a difference of 1.07–1.27% for α-tocopherol accumulation in lambs depending on the dietary dose and vitamin E source. Kim et al. [43] studied the effect of supra-nutritional vitamin E (35, 300 and 700 IU) supplementation for 14, 28 and 42 days before slaughter. They reported that vitamin E supplementation for 28 days before slaughter maximizes the *longissimus thoracis et. lumborum* muscle vitamin E concentration. An enhancement of vitamin E accumulation has been described when decreasing vitamin A levels from the diet in pigs [44,45] and lambs [36]. Selenium deficiency hinders vitamin E absorption and greater vitamin E concentrations in *longissimus*
*thoracis* muscle of pigs have been observed when organic selenium source was included in the diet [46]. Regarding genetics, most studies have reported no breed differences, and no relevant research was found providing genetic parameters for vitamin E in meat. Studies investigating changes in gene expression, protein abundance and the concentration of metabolites after vitamin E dietary inclusion are also limited. Factors such as vitamin E form and tissue type is known to affect gene expression. To better understand the underlying genetic mechanisms of the abosprtion and molecular mechanisms of vitamin E, one approach is to integrate omics technology to better understand its biological role and potential to be manipulated [47]. Currently, few studies have evaluated genetic parameters for vitamin E in muscle tissue. Ntawubizi et al. [48] described a heritability for α-tocopherol of 0.30 in plasma collected at slaughter from Pietrain (Landrace—Large White) pigs from two performance test stations.

B vitamins are water-soluble micronutrients which are best absorbed as a complex and play many important roles in the human body. Among other functions, thiamine (vitamin B_1_) enables conversion of blood glucose into biological energy, having a modulatory role in the acetylcholine neurotransmitter system. Riboflavin (vitamin B_2_) is involved in carbohydrate, protein and fat metabolism processes, as well as other biological mechanisms associated with fatty acid and iron metabolism and thyroid regulation. Forms of niacin (vitamin B_3_), such as nicotinamide adenine dinucleotide (NAD) and NAD phosphate (NADP) are included in a vast array of processes and enzymes involved in every aspect of peripheral and brain cell function. Pantothenic acid (vitamin B_5_) is part of co-enzyme A (vitamin B_6_). It is involved in protein metabolism, red blood cell metabolism, hemoglobin or neurotransmitter formation and maintaining blood glucose. Biotin (vitamin B_7_) is important for carbon dioxide fixation, carbohydrate and fat metabolism, and plays a key role in glucose metabolism and homeostasis. Folic acid (vitamin B_9_) acts as a co-enzyme for leucopoiesis, erythropoiesis and nucleoprotein synthesis, and is necessary for the synthesis and regeneration of monoamine neurotransmitters. Cobalamin (vitamin B_12_) is essential for the maturation of erythrocytes, cell growth and reproduction and the formation of myelin and nucleoproteins; low levels of cobalamin results in a functional folate deficiency [49]. In general, recommended amounts of these vitamins are achieved by humans through their diet and deficiency is more common in developing countries. However, in developed countries, the vegan sub-population is identified as at risk for deficiency, especially for cobalamin. This is because vitamin B_12_ is naturally present only in animal-derived products, which requires constant supplementation among individuals consuming vegan diets.

Regarding ruminants, B-complex vitamins are degraded in the rumen [50,51] and the nutritional requirements of the animal are usually met or exceeded with microbial fermentation in the rumen; therefore, dietary supplementation is not normally implemented by ruminant nutritionists. Vitamin B_12_ synthesis requires adequate levels of dietary cobalt. However, high levels of dietary starch reduce vitamin B_12_ levels in the rumen, leading to reduced vitamin B_12_ in meat from ruminants. A recent study suggested that the species of bacteria and bacterial consumption of vitamin B_12_ in the rumen may better represent overall levels, compared to bacterial production [52]. Duckett, Neel, Fontenot, and Clapham [34] compared high concentrate finishing systems to pasture finishing systems for Angus-cross steers and found nutrient potentials of 46.7 μg/100 g for B_1_ and 233 μg/100 g for B_2_. In addition to production systems, feed additives also have an impact on the nutrient potential of B vitamin concentration in meat. This has been observed using algae additives in pig feed, which led to nutritient potentials of 90 μg/100 g for B_6_ and 0.06 μg/100 g for B_12_ [53]. Regarding monogastric animals, diets are usually supplemented with B vitamins, and while additional increases have a small impact on muscle concentration of vitamins B_9_ and B_12_, vitamin B_2_ does not seem to respond to higher supplementation. To the best of our knowledge, no genetic studies stating heritability values for vitamin B concentrations in meat are currently available. Nevertheless, heritability values within the range of 0.23–0.45 were reported for B_12_ [54,55] and within the range of 0.31–0.52 for B_2_ [56] concentration in cow milk.

#### 3.1.2. Trace Elements

Meat is considered an important source of trace minerals for humans, having higher concentration compared to other foods [4] and a high content and bioavailability of copper, iron, phosphorus, magnesium and zinc [3,57].

Table 2 contains the compositional range of iron, copper, zinc, selenium and iodine in red meats including beef, pork and lamb, revealing high variation in concentrations of trace elements in red meat. Other studies have shown variability in trace element contents in meat, not only among species, but also among different muscles [58].

Although for most of the trace minerals, animal tissue concentrations are independent of intake, it has been suggested that dietary intervention [59] and genetic factors could be modified to manipulate the concentration of specific trace elements in meat with greater success than in other animal-derived food products, such as eggs or milk [60].

Copper is a cofactor for several enzymes and influences iron metabolism in the human body. Copper deficiency is not common in human diets; however, under certain conditions, such as celiac disease, lower intestinal absorption, anemia and osteoporosis could occur from copper deficiency. In addition, copper intake is regulated by the liver, and this regulation is different among livestock species. For this reason, while copper concentration is highly dependant on dietary sources, muscle concentration does not respond to direct supplementation and relatively low dietary copper concentrations can often be impaired [61,62,63,64]. Nevertheless, Ponnampalam et al. [65] found higher copper concentration in meat from lambs fed a high energy-high protein finishing diet compared with meat from lambs fed high energy–moderate protein or moderate energy–high protein finishing diets. Zhao et al. [66] reported higher concentrations of copper in organic pork than in conventional pork, partially due to the organic pigs having more time to exercise, which enhanced the capability of conserving this element. Regarding genetic manipulation, most studies looking at heritability of copper in muscle have reported no genetic variation. However, using Genome-Wide Association Study (GWAS) and a Bayesian approach in a small population of Nellore beef cattle, Tizioto et al. [67] reported moderate heritability estimates for copper content. Several studies have shown potential to manipulate plasma copper concentration [68]; however, this effect was not reflected in muscle content, which may be due to liver regulation.

Zinc plays an important role in the immune system of humans and zinc deficiency can impact growth in children and lead to various health issues in adults [69]. Zinc deficiency is often linked to low bioavailability due to interactions with phytic acid from plant dietary sources. The gastrointestinal-pancreatic control of zinc absorption makes it difficult to manipulate through dietary means. Most studies have reported little or no effect on muscle concentrations following zinc supplementation [66,70,71], especially in pigs. Interactions between trace elements should be considered, as high dietary zinc or zinc supplementation can interfere with copper and iron absorption [72]. Based on the literature found, genetic manipulation could have certain impacts in sheep, but little to no impact in beef cattle. The biological significance of this manipulation would most likely be minimal. In contrast, the effect of genetic line has been reported on zinc content in meat from lambs and pigs and is mainly associated with the muscle fiber type distribution [73,74].

Iron is an essential trace element in human diets and iron deficiency is an important health concern globally [75], affecting approximately 20% of the population and 50% of the population in less developed countries [76]. Dietary iron can be found in heme and non-heme form. The heme form has much greater bioavailability (20–30% absorption vs. 5–10% absorption for non-heme iron) and can only be obtained by consuming animal foods, such as red meats. Other influences on heme iron content include muscle fiber types, as fiber type proportion varies across different muscles. Absorption of heme iron is also relatively independent of other dietary ingredients, while absorption of non-heme iron can be influenced by meat composition. Heme iron is a component of hemoproteins, including myoglobin and, therefore, also influences the characteristic red color for meat. Contrastingly, free iron ions released from heme and ferritin are the main catalysts for lipid peroxidation of meat, and heme iron has been linked to increased risk of developing colorectal cancer by its implications in multiple processes in the intestine, such as DNA damage of epithelial cells and colonic hyperproliferation [77]. Hence, while red meats are recommended as an excellent alternative to decrease iron deficiency due to their heme iron content, this same factor may be the reason for some consumers to decrease their consumption of red meats. Regardless, understanding the potential manipulation of the concentration of iron content in meat is interesting from both quality and nutritional points of view. This has been well studied in red and white veal and has shown that using interventions to reduce iron content is easier than to increase iron content.

Iron absorption is controlled at the gastrointestinal level and, as previously mentioned, only a small portion of dietary iron is absorbed, making dietary manipulation difficult [78,79]. In fact, most studies that have shown a significant impact on increasing iron content in meat used extreme concentrations of dietary iron that surpassed the regulated limits allowed for use in animal diets [80]. Dimov et al. [81] found higher levels of iron in the meat of calves fed a silage-free finishing diet compared with calves fed a silage finishing diet, which could be attributed to the higher copper content found in the silage-free finishing diet. Other dietary ingredients, such as zinc supplements and green tea, have been reported to decrease iron content [82]. On the other hand, most studies have reported a relatively high heritability for both total iron and myoglobin content in red meat [67,83,84,85]. This is due to the link between muscle fiber type and iron content, in which meat containing more red muscle fibers have more iron compared to meat containing more white muscle fibers [70]. The few studies evaluating genetic parameters for muscle fiber type reported similar heritability values for total iron [82].

Another trace element is selenium, which is incorporated into selenoproteins and have several pleiotropic effects, including antioxidant and anti-inflammatory effects on the production of active thyroid hormone. Low selenium status in humans leads to increased risk of mortality, poor immune function and cognitive decline. In contrast, high selenium status or selenium supplementation is known to have positive effects, such as antiviral effects, improved male and female reproductive function and reduced risk of autoimmune thyroid disease [86]. Selenium-deficiency and elevated iodine together can have negative health impacts, such as enhanced autoimmune reactions and accelerated deterioration of thyroid function through oxidative stress [87]. These reasons highlight the need to provide sufficient selenium in the human diet, and one approach is through red meat consumption.

Selenium supplementation is common in livestock nutrition in order to meet the dietary requirements of animals. Its positive impact on meat quality as part of the antioxidant enzyme glutathione peroxidase also makes selenium supplementation interesting in regard to avoiding meat quality defects, such as white muscle disease in calves and lambs when low concentrations of selenium are present in the soil. Although no homeostatic regulation has been described for dietary selenium, bioavailability seems to be lower in ruminants, which may be due to transformation in the rumen. However, supplemented selenium can be deposited in meat tissue, and it is possible to produce selenium-enriched meat products. Several studies have reported the effect of the selenium source on absorption and deposition, with organic selenium, such as selenium-enriched yeast, increasing selenium supplement in muscle content at higher rates than inorganic selenium supplementation [88,89,90]. In terms of genetics, relatively high heritability has been reported for the selenium content in cattle [67], indicating that some variation of selenium content can be attributed to genetic variation in cattle. The latter study indicates the potential for genetic manipulation and the implementation of new animal breeding programs to improve the selenium concentration in muscle tissue and enhance nutritional attributes; however, in general, there remains a lack of studies evaluating genetic parameters of minerals and further research is needed on larger animal populations and in various breeds and species.

Iodine deficiency in humans is a severe condition that can lead to health issues including goiter, especially during childhood. It has been suggested that the high levels of iodine in animal-derived food products has decreased iodine deficiency in several countries [91]. The consumption of meat allows for sufficient dietary intake of iodine in humans, but maximum levels can actually be exceeded when overconsuming milk and eggs [91]. Iodine is directly absorbed in the intestine and regulated by the thyroid. Few studies show that increasing iodine content in meats is possible through dietary supplementation, in both ruminant and monogastric animals [92,93]. In lambs, a great increase has been observed in diets with high levels of seaweed. Similar to selenium, this may be the reason there are no available data regarding genetic parameters for this trace element.

### 3.2. Macronutrients

#### 3.2.1. Total Protein and Amino Acids

Proteins are essential macronutrients for human energy and nutrient requirements, as protein or amino acid deficiencies are known to cause severe health issues, especially in pregnant women, children [94] and the elderly. The consumption of indispensable (i.e., essential) amino acids, which cannot be synthesized in the human body, highlights the importance of consuming balanced protein sources from food. The amino acid balance and digestibility have been used to define protein quality in different food sources. Animal-derived products, including red meat, provide complete proteins. Animal protein is necessary, for instance, in situations where patients require high consumption of protein for tissue and musculoskeletal recovery [95,96]. Additionally, amino acid content plays an important role in the development of meat flavor compounds and sensory characteristics during cooking, which highlight the importance of amino acid composition for consumer acceptance of red meats.

With muscle protein being a functional tissue, studying the change in protein content of a muscle must consider changes in fat or moisture content which could therefore influence protein content. Thus, most of the studies reviewed have evaluated genetic and dietary approaches and their influence on changes in individual amino acid content. Descriptive studies have measured total protein of muscle tissue derived from red meat livestock species [3,97,98]; however, there is a lack of studies evaluating total protein or amino acid content in red meats and how these components can be modified to enhance its nutritional value using nutritional or genetic approaches [99]. Drazbo et al. [100] found that total protein in pork *longissimus dorsi* muscle was different when feeding a diet with protein and amino acid levels reduced by 15% relative to the standard levels. Specific pork loin essential amino acid composition was also different when supplemented with dietary additives like ginseng [101]. Muscle histidine and valine concentration have been observed to be lower when fed 5:1 and 10:1 PUFA ratio diets compared to 1:1 and 2.5:1 PUFA ratio (n-6:n-3) diets [102], suggesting lower concentrations of these amino acids are observed when fed a high saturated fatty acid profile. However, no effect on total protein was observed from diets with 2.0% supplemented palm oil and 0.5% or 1.0% CLA [103]. A study observed cattle fed grass silage had higher free amino acid levels compared to animals fed a concentrate diet, and many individual amino acid concentrations were also significantly different [104]. A study on lambs weaned at different ages revealed no difference in crude protein percent; however, looking at individual amino acids, essential amino acids were higher when weaning occurred at an earlier age [105], suggesting potential shifts in fiber type composition and therefore amino acid composition. Additionally, in the latter study, environmental factors such as production system or weaning system could have affected amino acid composition. A study using diet manipulation revealed protein percent in lamb leg was significantly impacted by feeding olive cake [106]; this suggests the increase in fat and decrease in moisture of the diet influences protein content.

Regarding genetic approaches to manipulate total protein or amino acid content in red meat, existing studies have revealed heritability of both total protein and individual amino acids ranges from low to high for beef and pork [107,108,109]. A prior study based on targeted single nucleotide polymorphisms (SNPs) and total significant SNPs revealed moderate to high heritability estimates of 0.42 and 0.26, respectively, for total protein content in pork [109]. In addition, pork breeds have also been compared to evaluate the different genetic backgrounds associated with total protein and amino acid profiles [110]. Studies have also estimated heritability of different amino acids in beef, revealing estimates of 0.34, 0.17, 0.66, 0.40 and 0.33 for alanine, glutamine, taurine, anserine and inosine, respectively [108]. Similarly, Ahlberg et al. [107] revealed high heritability estimates of total protein content in beef *semitendinosus* and *longissimus* muscles of 0.75 and 0.70, respectively. One study revealed that the *longissimus dorsi* mean amino acid content was significantly different between lamb breeds for arginine, glutamine and tyrosine [111]. The studies to date suggest a potential for manipulating individual amino acid concentrations content using diet, as well as genetic approaches, due to the moderate to high heritability of these components; however, further research is needed, specifically to evaluate the potential to manipulate total protein content.

#### 3.2.2. Total Fat and Fatty Acid Composition

Meat lipids continue to remain as the nutrient component with the highest potential for modification, in both content and composition, presenting opportunities for value added production and health promotion [112]. For instance, low-fat of n-3 enriched meats are considered functional foods for overweight individuals, since their consumption improves the body fat index, n-3 levels and the n-6:n-3 ratio, without impacting the Healthy Eating Index or intake levels of energy or other macronutrients [10].

Numerous studies have reported values for total fat, as well as different groups of fatty acids in meat. This area of research has been highly researched for the last 20 years and continues to attract attention from researchers in the area of animal and meat sciences (Table 3). The potential for manipulation of lipids is clear when the ranges in the literature were considered, with clear differences between monogastrics and ruminants. In monogastrics such as swine and poultry, meat fatty acid composition is reflective of their diets, whereas, in ruminants, dietary unsaturated fatty acids undergo extensive biohydrogenation by the rumen bacteria and are transformed into saturated fatty acids [113]. This phenomenon limits the ability to increase the content of these fatty acids in ruminant meats through feeding polyunsaturated fatty acids (PUFA) sources such as oilseeds and fish oil [114]. Conversely, during ruminal biohydrogenation of PUFA, several intermediates are produced, and a portion of them passes from the rumen and subsequently finds its way into meat after post-ruminal absorption. Specific biohydrogenation intermediates such as conjugated linoleic acids (CLA) and vaccenic acid (VA, *trans*-11 18:1) have been associated with several health benefits including anti-inflammatory and anti-diabetic effects [115].

Key targets for manipulation include increasing n-3 PUFA across species, and specifically in ruminants increasing contents of “healthy” PUFA biohydrogenation intermediates including CLA and VA [112,114]. In addition, a primary target has been to reduce saturated fatty acid (SFA) content as well as increasing levels of oleic acid (*cis* 9-18:1) [116]. Feeding grains, oilseeds, forages, grass or DDGS, among other feedstuffs, has a large impact on intramuscular fat (IMF) and the proportions of the different fatty acid groups. For example, meat from grass-fed ruminants tend to present lower IMF and higher proportions of n-3 PUFA, CLA and VA compared to concentrate-fed ruminants [27]. Feeding concentrate based diets, however, have been associated with decreased PUFA/SFA ratios, but, over the finishing period, there is increased conversion of SFA to monounsaturated fatty acids (MUFA), and relative rates are influenced by breed [117]. On the other hand, manipulation of PUFA biohydrogenation intermediates may have more to do with interactions between diet, rumen microbiology, and management than host genetics. Indeed, large increases in VA and CLA in steers has been linked to feeding management, for example, feeding a PUFA rich supplement (flaxseed co-extruded with peas) before feeding hay, instead of feeding a hay and supplement mix, has led to a substantial increase in VA and CLA [118,119]; these differences are related to shifts in the rumen microbial population [120].

While many studies have reported the potential for manipulation of VA in beef and lamb, it is important to look at the total amount of *trans* fatty acids and the relative proportions of *trans*-10-18:1 and *trans*-11-18:1, as *trans*-10 has been associated with detrimental effects on blood lipid profiles through upregulation of hepatic triacylglycerol and cholesterol synthesis [115,121]. Again, host genetics are not considered as a primary factor in the accumulation and proportions of *trans* fatty acids, but different forages, grass or vitamin supplementations can lead to shifts in *trans* fatty acid amounts and isomer proportions [27]. In beef and lamb, feeding sources of PUFA leads to a large number of biohydrogenation intermediates including several conjugated and non-conjugated 18:2 and 18:3 isomers for which the roles in the human body are still unclear [115]. CLA has also received more attention recently due to the potential health benefits of nitro-fatty acids in humans [122], but the role of host genetics in these processes is not yet clear. In fact, in ruminants, dietary ingredients and additives that modify the rumen microbiome may have a larger effect than direct supplementation of fatty acid supplements. On the other hand, dietary supplementation of PUFA has a large impact on pork fat composition, leading to large increases in PUFA and n-3, even long chain n-3, especially when using marine sources, such as fish oil and algae [28]. In cattle, however, some limitations exist regarding long chain PUFA deposition due to their preferential incorporation into phospholipids [123]. Further manipulation of pork IMF can also be achieved through altering lean deposition by reducing the protein or lysine content in diets (i.e., causing lean to fat repartitioning), or by adding CLA into pig diets [124].

The relationship between total fat content and relative proportions of fatty acids is important to consider, as higher IMF corresponds to lower relative PUFA content, due to the smaller contribution of membrane phospholipids [125]. For this reason, when reporting fatty acid profiles in meat, it is important to provide either total IMF or use mg per 100 g of meat as the unit, instead of the percentage of fatty acids in total fat. Similarly, consumers do not eat denuded muscles, but commercial cuts, which combine lean with seam fat and subcutaneous fat. Thus, while the manipulation of fatty acid profile in IMF may be more limited, it is possible to obtain a larger impact when considering changes in the whole primal, including all fat depots [126]. In general, dietary effects observed in IMF tend to have a larger impact on larger fat depots. This is important when trying to enhance the lipid profile in order to reach certain health claims. In fact, an alternative for fresh meat products, such as ground meat, could be accomplished by either supplementing the diets of a small percentage of the animals or selecting carcasses with a naturally higher concentration of certain beneficial fatty acids, and then mixing the fat from those carcasses with lean from the regular population. Manipulation of fatty acid profiles also has to take into consideration effects on meat and fat quality (taste, oxidative stability, fat softness, etc.), as enhancing the healthfulness of the fatty acid profile will be of limited value if overall quality is negatively affected. Thus, studies investigating manipulation of fatty acid profiles need to be linked with complimentary studies on meat or meat product quality, including sensory evaluation [112].

Inter- and intra-breed differences are well known in terms of total IMF variability, with very obvious cases of genetic groups with higher marbling [127,128], while populations selected for other traits correlated to total fat, such as lean meat yield, have seen a decrease in IMF as a negative side effect [129,130,131] Studies show a medium to high heritability not only for total IMF, but also for the majority of fatty acid groups which can be endogenously synthesized in both ruminants and monogastrics [132,133]. According to GWAS studies in different species, both total IMF and fatty acid composition in meat are influenced by key regulatory genes with major effects and multiple genes with smaller effects, and have shown moderate to high heritability estimates for IMF and low to medium heritability for specific fatty acids [134,135,136,137,138,139]. However, despite the potential to include IMF fatty acids in breeding programs, antagonistic genetic relationships with performance have usually minimized the emphasis on selection for these traits. A recent study [140] reported a series of genetic markers that could be used to manipulate IMF without impacting backfat thickness, opening new opportunities for animal selection. In terms of manipulating meat fatty acid composition, the influence of fatness on the lipid profile (decrease of relative proportion of PUFA with higher levels of IMF) must be taken into consideration [133]. Nevertheless, multiple studies have found SNPs for a number of candidate genes regulating intramuscular fatty acid metabolism [141]. Within the last few years, numerous studies have focused on alternatives to traditional genetic selection. This includes the use of transgenic animals (by nuclear transfer of modified DNA to an embryo) which can increase the endogenous production of certain beneficial fatty acids, such as omega-3 fatty acids [142,143,144,145,146]. This approach is also possible for feedstuffs, with crops being genetically modified to produce long-chain omega-3 fatty acids usually only available from marine sources [147]. Although these strategies present great potential for IMF and fatty acid manipulation, ethical and safety concerns still need to be addressed [148,149]. Moreover, consumer perception of genetically modified organisms and animal welfare could limit the large-scale implementation of these strategies [150].

## 4. Considerations

Enhancing the nutritional value of red meats continues to attract much attention from the scientific community and support from the industry. Research indicates great potential for the dietary manipulation of certain nutrients in red meats, such as vitamin E, selenium, total IMF, or fatty acid profile. In general, studies suggest greater potential for genetic selection for desirable IMF and fatty acid composition; genetic potential also exists for changing total protein when considering individual amino acids. Further research is needed to understand the genetic potential to manipulate iron content and therefore muscle fiber type composition. Fat content and lipid profile represented the fraction with the highest potential for manipulation either through diet or genetic selection [112]. However, most studies have used approaches that independently evaluate the impact of either genetics or nutritional strategies, and few studies have explored the interactions between these two major factors [151,152,153]. Furthermore, recent research on microbiome manipulation have shown an impact on meat composition [154]. Holistic approaches, such as livestock precision farming, systems biology, livestock phenomics, and nutrigenomics, have the ability to integrate genetic, environmental and phenotypic information, leading to better understanding of the biological system as a whole and unlocking the true potential for manipulating meat nutritional attributes [155,156].

Establishing the justification to modify the nutritional content of red meat and understanding its consequences should also be considered due to ethical concerns. Among the reasons commonly described to justify enhancing meat nutritional value, the importance of balancing currently deficient diets in developed countries, especially for populations at risk, should be mentioned, as well as the need for a more complete nutrient-dense product for the diets of developing countries [157,158]. Furthermore, new research findings and dietary recommendations from public and private institutions will continue to shape our understanding of the impact of different foods on human health; therefore, providing alternative approaches to modify the nutritional value of meat will allow the industry to address future challenges in this area [159]. However, the main justification identified in research studies is the production of value-added differentiated products [160,161]. Other sectors based on animal products, such as the egg and dairy industry, have developed strong marketplace differentiation based on nutritional enhancement claims [158]. With the exception of further processed meats, this strategy has reached lower success in the meat sector compared to other industries. While some fresh meats are using enhanced nutritional value as their differentiation strategy (e.g., omega-3 pork), credence attributes have become the most common form of differentiation to appeal to consumers with higher standards for aspects such as animal welfare, environmental concerns or the impact of livestock production on antimicrobial resistance [162,163]. Ethical aspects and sustainability concerns could be raised regarding the use of highly valuable feedstuffs, such as feed ingredients sourced from marine resources, for animal feed [164]. Additionally, changes in diet or genetic selection should consider potential impacts on animal welfare, as well as consumer perception of certain practices. As an example, the current perception from a large part of the population regarding genetically engineered foods could negate any commercial benefit from the use of transgenic animals or feedstock to enhance the nutritional value of meat [165]. In addition to the consideration of societal perspectives, limitations when using genetic selection should be considered, including the selection for desirable traits which may be negatively or positively correlated with other undesirable traits. Furthermore, although nutritional enhancement claims can be attractive for some consumers, it is well known that modifying meat composition can lead to changes in appearance, firmness, shelf-life and palatability; all of which could influence the consumer’s acceptance [166,167]. Surveys have shown that, while consumers may choose to consume chicken for its perception as a healthy alternative, the drivers leading to the purchase of red meats are mainly related to the eating experience [163]. Since palatability will ultimately determine consumer satisfaction [168], research is needed to evaluate quality attributes and how they are affected when using different strategies to modify the nutritional value of meat.

## 5. Conclusions

Traditional and novel omics research indicates the potential to manipulate fresh meat composition and therefore enhance its nutritional value. Enhancing the nutritional value of meat could be maximized by combining nutritional strategies and genetic selection, opening opportunities to develop added-value products. Certain limitations need to be considered due to complex metabolic processes and the influence of genetics on certain nutrients. Furthermore, to achieve sustainable animal production, holistic consideration of dietary and genetic strategies and its effect on animal welfare, environmental impact, product quality, and consumer perception must be considered. Traditional red meat will continue to serve as a nutrient dense food which is widely consumed due to the co-evolution of positive human sensory perceptions and thus will continue to provide health benefits to a large percentage of the world population.

## Figures and Tables

**Table 1 foods-10-00872-t001:** Nutrient range of vitamins in red meats.

Vitamins	Beef	Pork	Lamb
Vitamin A (μg/100 g)	5.00–11.5	2.00–6.10	7.80–8.60
Vitamin E (mg/100 g)	0.03–1.10	0.01–0.86	0.08–1.20
B_2_ (mg/100 g)	0.09–0.80	0.05–1.23	0.11–0.25
B_12_ (μg/100 g)	0.40–3.10	0.30–1.10	0.60–2.50

**Table 2 foods-10-00872-t002:** Trace element nutrient composition range in red meat.

Trace Elements	Beef	Pork	Lamb
Copper (mg/100 g)	0.04–1.40	0.03–0.59	0.03–0.13
Iron (mg/100 g)	1.00–7.80	0.30–3.00	1.10–3.60
Zinc (mg/100 g)	2.30–7.70	0.40–5.00	2.10–9.40
Selenium (mg/100 g)	0.40–10.8	0.05–1.23	0.30–35.0
Iodine (μg/100 g)	0.20–20.0	0.40–17.0	30.0–46.0

**Table 3 foods-10-00872-t003:** Nutrient range of fatty acids in red meats.

Lipids	Beef	Pork	Lamb
Total IMF (g/100 g meat)	0.60–26.90	1.60–17.00	2.50–18.10
SFA (g/100 g IMF)	33.70–49.10	32.80–41.00	46.20–50.40
MUFA (g/100 g IMF)	24.70–56.10	39.60–49.10	32.10–45.30
*Trans* (g/100 g IMF)	1.50–4.00	–	3.00–6.30
PUFA (g/100 g IMF)	2.80–29.00	3.80–26.20	3.60–8.10
n-3 (g/100 g IMF)	0.38–10.40	1.20–13.40	1.50–3.50
LC n-3 (g/100 g IMF)	0.25–4.90	0.19–1.90	0.77–1.40
n-6 (g/100 g IMF)	2.80–20.20	8.70–12.80	2.10–4.60
CLA (g/100 g IMF)	0.10–1.80	0.04–3.60	0.57–1.50

## Data Availability

This study did not report any data.

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
