# Peer review of "Enhancing the Nutritional Value of Red Meat through Genetic and Feeding Strategies"

_foods, 2021, doi:10.3390/foods10040872_

Round 1

Reviewer 1 Report

I have really appreciated the study conducted by Juárez et al. It addresses an emergent topic and reveals interesting results. However, there are some minor questions that I would like the authors clarify.

I think that the inclusion of a table that clearly shows the difference between the bioavailability of the main red meat nutrients in comparison with other food should be appropriate.

With the increase of vegetarian habits among the world population, are there any serious problems for these people to not include meat/red meat products in their diets?

Some figures need to be provided and also more tables to summarize all the information presented in the text.

Further investigations need to be pointed out at the end of conclusions. This section needs to be expanded

Author Response

Point 1: I have really appreciated the study conducted by Juárez et al. It addresses an emergent topic and reveals interesting results. However, there are some minor questions that I would like the authors clarify.

Response 1: Thank you for your feedback which has helped improve the manuscript.

Point 2: I think that the inclusion of a table that clearly shows the difference between the bioavailability of the main red meat nutrients in comparison with other food should be appropriate.

Response 2: The aim of this review was to report the past and current knowledge of nutritional and genetic approaches used to manipulate the nutrient composition of red meats and discuss the potential of integration of these approaches, as well as novel approaches. Regarding the bioavailability of red meat nutrients, a comparison of nutrient density and nutritional value of different red meats and non-meat foods has been previously done in a Review by Bohrer (2017). With that said, the authors believe this addition would be repetitive to Bohrer (2017) and outside the scope of this review.

Bohrer, B.M. Review: Nutrient density and nutritional value of meat products and non-meat foods high in protein. Trends Food Sci. Technol. 2017,65, 103–112. doi:10.1016/j.tifs.2017.04.016.4.

Furthermore, the authors have included a statement in the introduction to clarify this.

Page ?; L??

“The nutrient density and nutrient value of meat products has been reviewed and measured, showing the highly bioavailable and nutrient dense quality of beef, pork, and lamb compared to other non-meat food products.”

Point 3: With the increase of vegetarian habits among the world population, are there any serious problems for these people to not include meat/red meat products in their diets?

Response 3: Thank you for your comment, we have included a sentence and reference in the introduction.

Page ?; L??

“Furthermore, studies are emerging showing the negative health effects of omitting red meat in the human diet such as lower bone mineral density and higher bone fracture rates.”

Iguacel, I.; Miguel-Berges, M.; Gómez-Bruton, A.; Moreno, L.; Julián, C. Veganism, vegetarianism, bone mineral density, and fracture risk: a systematic review and meta-analysis. Nutr. Rev. 2019, 77, 1–18. doi: 10.1093/nutrit/nuy045.

Point 4: Some figures need to be provided and also more tables to summarize all the information presented in the text.

Response 4: We believe the appropriate tables are provided to inform readers on the genetic or nutrient potential to manipulate the red meat nutrients of interest in the review.

Point 5: Further investigations need to be pointed out at the end of conclusions. This section needs to be expanded

Response 5: Thank you for your comment. The conclusion has been restructured to be more concise and suggest further investigations.

Page ?; L??

“Traditional and novel -omics research indicate the potential to manipulate fresh meat composition and therefore enhance its nutritional value. Enhancing the nutritional value of meat could be maximized by combining nutritional strategies and genetic selection, opening opportunities to develop added-value products. Certain limitations need to be considered due to complex metabolic processes and the influence of genetics on certain nutrients. Furthermore, to achieve sustainable animal production, holistic consideration of dietary and genetic strategies and its effect on animal welfare, environmental impact, product quality, and consumer perception must be considered. Traditional red meats will continue to serve as a nutrient dense food which is widely consumed due to the co-evolution of positive human sensory perceptions and thus providing health benefits to a large percentage of the world population.”

Reviewer 2 Report

Review of paper by Juárez et al., Enhancing the nutritional value of red meat through genetic and feeding strategies. (Manuscript number: foods-1167898).

In the review, the Authors briefly – by using the most important literature sources - summarized the possibility of influencing the nutritional value of red meats through feeding and their genetic improvement. It is a well-written review. The conclusions are well-founded.

Some remarks (suggestions)

L43-61: I suggest writing briefly about the results of those studies that have been achieved in reducing greenhouse gas production. This is a very important factor in “protecting” red meat.

L27-42: It should be emphasized that a negative effect of red meats has been shown mainly in the case of large amounts of consumption and processed red meats. However, for pregnant and lactating women and for the elderly, it is very important to consume the required amount of red meat.

Perhaps, it should be emphasized even more that generally it is not based on scientific results and meta-analyzes that red meat consumption has a negative effect on people, but mainly it is written by less scientifically prepared food advisers, bloggers, etc., who directly influence the choices of consumers, especially young people through blogs, websites, newspapers, etc., which contain a lot of unscientific and false information.

It could be useful to show the composition of some white meats in the tables. Only to show the differences between them without a detailed analysis.

I suggest writing the n-6 / n-3 ratios in Table 3.

Somewhere at the beginning of the feeding section, it should be briefly summarized what is basically the difference between ruminant and monogastric animals and the meat of grazing (grass-fed) and concentrate-feed animals. Although Authors mention this in several places, however, it would be useful to write some information about it also at the beginning. This is only a suggestion.

Author Response

Point 1: In the review, the Authors briefly – by using the most important literature sources - summarized the possibility of influencing the nutritional value of red meats through feeding and their genetic improvement. It is a well-written review. The conclusions are well-founded.

Response 1: Thank you for your helpful comments; we have applied your suggestions or provided clarification.

Some remarks (suggestions)

Point 2: L43-61: I suggest writing briefly about the results of those studies that have been achieved in reducing greenhouse gas production. This is a very important factor in “protecting” red meat.

Response 2:

That is an important consideration and we have included your point in the introduction.

 Page L

“It is known that diets high in refined foods, sugars, oils and meats are expected to contribute up to 80% of the increase in GHG originating from food production and global land clearing.”

Point 3: L27-42: It should be emphasized that a negative effect of red meats has been shown mainly in the case of large amounts of consumption and processed red meats. However, for pregnant and lactating women and for the elderly, it is very important to consume the required amount of red meat.

Response 3:

That is an important consideration and we have included your point in the introduction.

Page L

“In support, studies have shown the importance of red meat nutrients for the pregnant and aging population for maintaining healthy vitamin and mineral status and skeletal muscle mass, respectively.”

Point 4: Perhaps, it should be emphasized even more that generally it is not based on scientific results and meta-analyzes that red meat consumption has a negative effect on people, but mainly it is written by less scientifically prepared food advisers, bloggers, etc., who directly influence the choices of consumers, especially young people through blogs, websites, newspapers, etc., which contain a lot of unscientific and false information.

Response 4: We agree with your comments. We have adjusted the introduction and hope the second paragraph regarding negative attention towards red meat consumption has emphasized the need to more critically assess both scientific and popular press statements about red meat consumption.

Point 5: It could be useful to show the composition of some white meats in the tables. Only to show the differences between them without a detailed analysis.

Response 5: The composition of white meats would be important to consider, however the focus of the review and discussion is on red meats and not white meats, and adding this information may be misleading. The authors believe adding this information would be outside the scope of this review.

Point 6: I suggest writing the n-6 / n-3 ratios in Table 3.

Response 6: Although the n-6/n-3 ratio is an important consideration, this value is more informative when considering human health and dietary balance and the review was focused on the manipulation and not the dietary needs of the population.

Point 7: Somewhere at the beginning of the feeding section, it should be briefly summarized what is basically the difference between ruminant and monogastric animals and the meat of grazing (grass-fed) and concentrate-feed animals. Although Authors mention this in several places, however, it would be useful to write some information about it also at the beginning. This is only a suggestion.

Response 7: We agree on the importance of clarifying this in the review. We have rewritten this part of the introduction to provide some clarity but did not include details to the different feeding strategies as there was a wide range of diet and feeding regimes covered in the following sections.

Page L

“Variability in the nutrient composition of red meats is known to exist due to differences in animal species (ruminant and monogastric) and breeds, as well as different strategies for feeding and rearing livestock, which has allowed for opportunities to manipulate the nutritional value of meat.”

Reviewer 3 Report

please see attached

Author Response

Response to Reviewer 3 comments were included directly in the PDF. The following changes were made:

Response: Thank you for your helpful comments. We have made revised the manuscripts to your suggestions. Please review the PDF comments or the updated manuscript to see our changes and comments.

This manuscript is a resubmission of an earlier submission. The following is a list of the peer review reports and author responses from that submission.

Round 1

Reviewer 1 Report

I have really appreciated to review the work conducted by Juárez et al. It addresses an emergent topic and reveals interesting results. However, there some questions that I would like the authors clarify before it can be considered for publication.

“vitamin” is not an adequate keyword for this manuscript.

The authors should take in consideration and discuss the most recent dietary guidelines around the world according this the topic of the present manuscript:

Fernandez, M.L.; Raheem, D.; Ramos, F.; Carrascosa, C.; Saraiva, A.; Raposo, A. Highlights of Current Dietary Guidelines in Five Continents. Int. J. Environ. Res. Public Health 202118, 2814. https://doi.org/10.3390/ijerph18062814

It should be interesting to clearly state and deepen analyze the difference between the bioavailability of the main red meat nutrients in comparison with other food. I also think that the inclusion of a table should be adequate.

With the increase of vegetarian habits among the world population, are there any serious problems for these people to not include meat/red meat products in their diets?

How to decrease the environmental impacts correlated with reat meat production? It would be interesting to provide some information about this.

The inconveniences of genetic manipulations need to be stated out by the authors.

Some figures need to be provided and also more tables to summarize all the information presented in the text.

Further investigations need to be pointed out at the end of conclusions.